# Integrating Modular Pipelines with End-to-End Learning: A Hybrid Approach for Robust and Reliable Autonomous Driving Systems

**DOI:** 10.3390/s24072097

**Published:** 2024-03-25

**Authors:** Luis Alberto Rosero, Iago Pachêco Gomes, Júnior Anderson Rodrigues da Silva, Carlos André Przewodowski, Denis Fernando Wolf, Fernando Santos Osório

**Affiliations:** Institute of Mathematics and Computer Science, University of São Paulo, Ave. Trabalhador São-Carlense, 400, São Carlos 13564-002, SP, Brazil; iagogomes@usp.br (I.P.G.); junior.anderson@usp.br (J.A.R.d.S.); carlos.andre.filho@usp.br (C.A.P.); denis@icmc.usp.br (D.F.W.)

**Keywords:** autonomous driving, hybrid architecture, modular, end-to-end, path planning, CARLA simulator

## Abstract

Autonomous driving navigation relies on diverse approaches, each with advantages and limitations depending on various factors. For HD maps, modular systems excel, while end-to-end methods dominate mapless scenarios. However, few leverage the strengths of both. This paper innovates by proposing a hybrid architecture that seamlessly integrates modular perception and control modules with data-driven path planning. This innovative design leverages the strengths of both approaches, enabling a clear understanding and debugging of individual components while simultaneously harnessing the learning power of end-to-end approaches. Our proposed architecture achieved first and second place in the 2023 CARLA Autonomous Driving Challenge’s SENSORS and MAP tracks, respectively. These results demonstrate the architecture’s effectiveness in both map-based and mapless navigation. We achieved a driving score of 41.56 and the highest route completion of 86.03 in the MAP track of the CARLA Challenge leaderboard 1, and driving scores of 35.36 and 1.23 in the CARLA Challenge SENSOR track with route completions of 85.01 and 9.55, for, respectively, leaderboard 1 and 2. The results of leaderboard 2 raised the hybrid architecture to the first position, winning the edition of the 2023 CARLA Autonomous Driving Competition.

## 1. Introduction

There are different methodologies for developing an autonomous system, which involves software components and algorithms from fields such as machine learning, computer vision, decision theory, probability theory, and more. This mix of components adds complexity to both the development and evaluation process [1]. Figure 1 illustrates the main difference between the three approaches for architecture, being modular, end-to-end, and hybrid architectures.

The standard method employs modular pipelines and has proven effective in scenarios with access to detailed high-definition (HD) maps or dense waypoints. This approach, widely adopted by both companies and research groups [2], decomposes the navigation problem into specific tasks such as localization, object detection, tracking, prediction, decision-making, path planning, and control [3,4,5]. The advantage of this architecture lies in its interpretability, enabling a comprehensive evaluation of each component. However, the extensive coupling of numerous components increases the risk of error propagation, leading to increased complexity in maintaining the entire architecture and a rise in associated costs.

Conversely, the end-to-end approach aims to directly map sensor input to driving actions, bypassing explicit task decomposition [6]. This model leverages deep learning techniques, employing neural networks to learn complex mappings from raw sensor data to steering, braking, and throttle commands. While this method simplifies the system architecture and reduces the need for manual feature engineering, it often requires vast amounts of training data and lacks transparency in decision-making processes.

Autonomous navigation in complex environments has progressed with modular and end-to-end approaches. However, each method has limitations, prompting the development of a hybrid model that combines their strengths. Module-based systems, with their numerous components and inflexibility, may struggle to adapt to diverse and unpredictable situations, particularly when reliant on precise map data. On the other hand, end-to-end models, though effective, often operate as black-box systems, making it challenging to interpret their decision-making processes. Additionally, these models demand extensive training data, which can be impractical and costly for covering a wide range of driving scenarios. The hybrid model aims to address these issues by integrating the advantages of both modular and end-to-end approaches. This approach seeks to strike a balance between the interpretability of modular systems (with a better performance when considering maps, and more focused on deliberative tasks) and the learning capabilities of end-to-end models (with a better performance when considering complex sensorial input data, and more focused on reactive tasks), offering potential improvements in system robustness, combining deliberative and reactive behaviors, flexibility, adaptability, and overall performance.

Another challenge in autonomous driving is evaluating the autonomous driving architecture, whether it adopts a modular, end-to-end, or hybrid methodology. Kalra and Paddock [7], Koopman and Wagner [8], and Huang et al. [9] suggest that to comprehensively assess an autonomous system, it is important to combine real-road and simulation tests. In this regard, simulators offer advantages by creating repeatable scenarios for component performance assessments. They simulate diverse driving situations with realistic dynamics, including weather conditions, sensor malfunctions, traffic violations, hazardous events, traffic jams, and crowded streets. Simulations also serve as effective benchmarks, enabling the evaluation of different system approaches under the same conditions for comparative analysis.

This paper introduces an autonomous driving software architecture employing a hybrid methodology, integrating modular perception and control modules with data-driven path planning. The architecture was designed and evaluated during the 2023 CARLA Autonomous Driving Challenge (CADCH) (Available at: https://leaderboard.carla.org/challenge (accessed on 5 February 2024), achieving first and second place in the SENSORS and MAP tracks, respectively (Available at: https://leaderboard.carla.org/leaderboard (accessed on 5 February 2024)). The 2023 CADCH represents the fifth edition of the autonomous driving competition set in urban simulated environments using the CARLA, featuring diverse urban scenarios. Our involvement spans multiple editions, and we secured victory in the inaugural challenge through a modular and vision-based navigation stack [10]. Therefore, the primary contributions of this paper are the following.
A hybrid software architecture for autonomous vehicles, combining modular perception and control modules with data-driven path planning;A comprehensive comparison between modular and hybrid software architectures through the simulation of urban scenarios;Evaluation of autonomous driving performance in diverse and hazardous traffic events within urban environments.

The remainder of the paper is organized as follows: Section 2 provides a critical overview of related-works; Section 3 describes the modular software architecture developed in the competition; Section 4 presents the hybrid architecture approach; Section 5 discusses the results of the competition and other experiments; finally, Section 6 addresses the final remarks and suggests some future work.

## 2. Related Works

As mentioned before, there are different approaches to developing autonomous systems, depending on the technologies used and how the components are structured. This section offers a brief and critical review of modular, end-to-end, and hybrid software architectures. Table 1 summarizes the related works.

### 2.1. Modular Navigation Architecture

A modular architecture typically organizes software components in a hierarchical manner based on specific criteria. Each group, referred to as a layer, operates at a distinct level of abstraction and provides services to its adjacent layers. This structure follows a descending order of abstraction, where higher-level layers handle more abstract tasks, while lower-level layers manage finer controls in the architecture. For example, following the hierarchical navigation stack proposed by Paden et al. [27], the initial layer is responsible for road and lane-level route planning, determining the roads and lanes the vehicle must follow to reach its destination. Subsequently, a behavior layer makes tactical decisions for the vehicle during navigation, such as interactions with other traffic participants, adherence to traffic rules, and high-level maneuver choices (e.g., lane following, lane change, U-turn, overtaking, and emergency stop). In addition to the route, this layer also receives perception information, including obstacle position and velocity and traffic light status. Once the behavior is determined, a motion planning layer calculates short-term, feasible, and collision-free trajectories that are translated into low-level commands, such as throttle, brake, and steering, by low-level controllers within the control layer [28].

This design pattern is widely utilized in autonomous systems and has demonstrated notable success in both industrial and research vehicle applications [2]. Key studies typically adopt similar layers, including sensing, perception, planning, control, and human–machine interfaces as fundamental components [4,11,12,13,14]. However, there are variations, such as communication between vehicles (e.g., vehicle-to-anything—V2X) [11]; health management systems focusing on hardware and software component monitoring, diagnosis, prognosis, and fault recovery [4,12,14]; behavior or mission planning [4,13,14]; and mapping strategy [4,11,13]. In the latter case, Wei et al. proposed an alternative to the hierarchical navigation stack. This alternative organizes components of the behavior and control layers in parallel order, based on the functioning of the ADAS (Advanced Driver Assistance) system. According to the authors, this approach enhances the flexibility of the autonomous system, enabling it to operate at a higher frequency compared to alternative methods.

Nevertheless, the parallel design faces challenges in coordinating components during complex tasks and maneuvers due to asynchronous communication. Additionally, both parallel and hierarchical approaches share issues related to error propagation between components and the intricate management of components with the increase in vehicle autonomy. This occurs because, as autonomy increases, the vehicle also performs more tasks (e.g., maneuvers) and encounters a broader range of traffic scenarios. Therefore, the number of components adversely affects the system’s performance. To minimize error propagation across components, one approach is to use components that handle uncertainties effectively in lower levels of the navigation stack [29]. This includes employing robust control, decision-making that addresses partially observable issues, and trajectory planning that is aware of occlusions and uncertainties in obstacle positions and speeds. Additionally, incorporating fault detection, diagnosis, prognosis algorithms, and health management systems for architectural components is crucial [30]. While these approaches do partially alleviate the impact of error propagation and other issues linked to modular architecture, they result in increased complexity and maintenance costs for the architecture.

### 2.2. End-to-End Autonomous Driving

End-to-end is a navigation approach where neural networks and deep learning models are trained to map sensory input (e.g., images or point clouds) to control outputs (e.g., steering, throttle, brake) or intermediate outputs (e.g., trajectory segment). This eliminates the need for manual feature tuning in modular navigation pipelines. The advantage lies in leveraging deep learning generalization to simplify and enhance the adaptability of navigation stacks across different traffic scenarios. There are various approaches to classifying end-to-end models, ranging from the degree of the deep learning model’s involvement in tasks to the technology applied. In the former category, methods range from pure end-to-end architectures, where the deep learning models handle the entire mapping and decision-making process, to hybrid approaches that integrate different algorithms, such as probabilistic models, control theory, fuzzy inference systems, etc. The latter category divides models based on techniques, such as imitation learning and reinforcement learning.

In addition to the presented taxonomy, studies on end-to-end navigation also focus on input representation aspects and model design. This includes considerations in the number of cameras (e.g., single or multi-camera setups) [15,16,17], methods for 3D data representation (e.g., point cloud or Bird’s Eye View images) [15,16,18,20], sensor fusion and multimodality (e.g., different sensors and feature fusion methods) [19,20,21], interaction with traffic agents (e.g., interaction graphs or grid maps) [15,20], deep learning technologies (e.g., transformers, graph neural networks, deep reinforcement learning, attention mechanisms, generative models, etc.) [15,16,20,21], decision-making within the network (e.g., high-level commands input or inference) [17,18], and the accuracy or feasibility of the output (e.g., using standard controllers to estimate final outputs or filtering the output of the deep learning model) [15,18,21].

In summary, end-to-end models primarily rely on RGB images and LiDAR-generated point clouds, represented in 3D as points, voxels, or bird’s eye view (BEV) images. While the ResNet network is commonly used for feature extraction from images and BEV [16,17,18,20,21], some studies also explore the use of specialized deep learning models for 3D data, such as PointPillars [15] and VectorNet [20]. Two significant challenges for deep learning models include multimodality fusion and how to handle tactical decisions within the network (or when incorporating decisions from external sources). In the former case, early-fusion and middle-fusion approaches are noteworthy [19], they often involve attention mechanisms or concatenation of feature vectors. In the latter case, tactical decisions (i.e., high-level commands) can be treated as an input modality [17] or a conditional variable [18,19], particularly in approaches exploring multiple-expert designs. However, both pure and hybrid end-to-end navigation methods still face challenges related to the lack of transparency and explainability in decision-making and the requirement for extensive training data.

### 2.3. Data-Driven Path Planning

Autonomous vehicles rely on path planning algorithms to navigate through dynamic and complex environments. Data-driven approaches have gained prominence in recent years, representing a shift from traditional rule-based methods [31]. In data-driven path planning, algorithms leverage machine learning techniques to learn collision-free paths from large datasets [32]. These datasets typically include information from various sensors, historical driving experiences, and diverse environmental conditions. Similar to end-to-end navigation architectures, data-driven path planning also inherits the adaptability features from deep learning models, which make them able to plan under diverse road geometry and traffic scenarios.

Techniques for data-driven path planning typically emphasize the representation of spatial and temporal features. However, to address the challenges of dynamic driving scenarios, they also consider the representation of traffic rules, interaction among traffic participants, output trajectory smoothness and comfort, high-level commands (e.g., maneuvers), and variations in road geometry. Spatial features are commonly derived from frontal camera images or bird’s eye view (BEV) projections, using CNN-based networks for feature embedding [23,24,25,26]. Some works also use the historical trajectory of the ego-vehicle and surrounding agents [22]. Temporal features are traditionally addressed by recurrent neural networks (e.g., gated recurrent unit—GRU and long short-term memory—LSTM) [23,25,26], although recent studies have explored the application of Transformer networks [22]. Semantic and abstract data, such as traffic rules and high-level commands, are integrated as feature vectors or conditional variables [26]. Finally, ensuring trajectory smoothness typically involves the application of a post-processing algorithm or the penalization term in the loss function [26]. Nevertheless, methods employing machine learning for path planning face challenges in terms of transparency and explainability. Moreover, there is room for improvement in addressing global planning, high-level commands, managing dangerous and unexpected driving scenarios, and ensuring dynamic and kinematic feasibility of planned trajectories.

Sensor fusion significantly improves the reliability of perception and data-driven path planning in autonomous systems [33]. Integrating information from various sensors enhances the system’s understanding of its surroundings [34]. Different sensors, such as cameras, LiDAR, and radar each have unique strengths and weaknesses. Combining their data results in a more robust representation of the environment. For instance, cameras offer rich visual information, while LiDAR provides precise distance measurements [35]. This integration ensures a more accurate perception, enabling the system to generate safer trajectories, since knowing the precise positions of obstacle, road boundaries, and other features enhances the path planning system’s capability to navigate complex scenarios.

In this context, this paper introduces a hybrid autonomous vehicle architecture that integrates modular pipelines with data-driven path planning, offering a comprehensive comparison of these approaches. This architecture, developed and evaluated in the 2023 CARLA Autonomous Driving Challenge (2023-CADCH), secured first and second place in the SENSORS and MAP tracks, respectively. By combining modular perception and control components, it delivers reliable information to the data-driven path planning, ensuring the generation of kinematically feasible trajectories. Additionally, an early-fusion approach enhances the spatial and semantic representation of the environment in the data-driven path planning through the fusion of LiDAR Bird’s Eye View images, Stereo Camera projections, and high-level commands. The results demonstrate the network’s generalization capabilities with real-time inference, highlighting the architecture’s reliability.

## 3. Proposed Modular Pipeline

An autonomous system requires several components and its architectural design provides an abstract view of the system operation and organization. In a layered architecture, the components have public and well-defined communication interfaces through which they exchange information with other components. This characteristic enables the definition of a common architecture for all tracks in this challenge, through adjustments of a few components for the maintenance of the same communication interface. This strategy reduces the time spent on the development of the agents, and enables the evaluation of the autonomous navigation performance with different sensors and algorithms for a specific task.

Figure 2 shows the general software architecture designed for all agents of the Laboratório de Robótica Móvel (LRM) team in the 2023 CARLA Autonomous Driving Challenge. The name “*CaRINA Agent*” is used to refer to this architecture in the rest of this paper. The layers of the architecture are *sensing*, *perception*, *map*, *risk assessment*, *navigation*, *control*, and *vehicle*. Robotic framework ROS (robotic operating system) supported the communication interface between components with the Publish/Subscribe pattern for message passing [36].

### 3.1. Mapping and Path Planning

A map is an essential component enabling the autonomous vehicle to execute its tasks safely and efficiently, storing diverse information, beneficial for various components of the autonomous system [37]. This includes the road geometry description for path planning and the topological representation of roads and intersections, commonly referred to as the road network, used for route planning. In addition to static object positions, navigable areas, positions of traffic signs and lights, traffic rules, and semantic information related to the road. In this architecture, we employed the OpenDRIVE [38,39] map standard to assist the navigation and perception components.

#### 3.1.1. OpenDRIVE

The OpenDRIVE is an open format to describe road networks, using XML version 1.0, which is able to represent the road geometry as well as the context information of roads that may influence the behavior of vehicles driving in it, such as traffic signs, traffic lights, and the type of roads and lanes (e.g., highway and sidewalk) [39,40]. The format description is built on a hierarchical structure with four main elements: *header*; *road*; *junction*; and *controller*. Figure 3 shows the visualization of the OpenDRIVE map after being parsed by the *map manager* in the architecture.

The *header* is the first element of the description and holds the metadata of the map, such as the name or a geographic reference for transformations between the Cartesian and Geodesic coordinate systems. The *road* element encompasses the geometry description and additional properties (e.g., elevation profile, lanes, and traffic signs). The properties, such as traffic signs, are placed with respect to the distance to the initial point of the lane, using a local coordinate system. Besides that, *roads* can connect directly or through intersections using the *junction* component, preventing ambiguities in connections. Lastly, a *controller* is employed for signalized junctions or other road elements imposing control on the vehicle.

#### 3.1.2. Path Planning

Path planning is responsible for determining a feasible and optimal path from a starting point to a destination in a given environment. This planned path considers factors such as road geometry, static and dynamic obstacles, vehicle physical constraints, and criteria like time, speed, fuel efficiency, or safety. Various algorithms, including rule-based, gradient-based, graph-based, and optimization methods, can perform these tasks. In addition, recent studies have explored the integration of machine learning into path planning. Moreover, the task can be divided into two steps: global planning and local planning. Global planning estimates a reference path, considering static features (e.g., road geometry and static obstacles) and the intended route. Local planning adjusts this global reference path based on dynamic variables in the traffic scene, such as dynamic obstacles.

In the modular navigation pipeline strategy, *global reference planning* incorporates a local speed profile based on the dynamic scene, along with local lane-change planning due to traffic events. The *global reference planning* involves sparse waypoints representing the intended route. The simulation supplies sparse waypoints at the start of each route. The goal is for the vehicle to navigate through each waypoint, encountering various traffic scenarios configured by the competition between them. Subsequently, lane-level localization determines the corresponding lane and road ID for each waypoint using the OpenDRIVE map manager. This information enables the system to estimate a lane-level route, identifying the roads and lanes the vehicle must traverse to reach its destination. Finally, segments of the reference path, each spanning 50 m in leaderboard 1 and each 100 m in leaderboard 2 are published in the ROS ecosystem based on the vehicle’s speed and current position.

### 3.2. Perception

The autonomous driving systems proposed in this paper rely on two types of sensors for robust environmental perception: a stereo camera and a LiDAR sensor. Our proposed perception system, depicted in Figure 4, adopts a multi-sensor fusion approach to achieve accurate and robust object detection in 2D/3D images and point clouds. This approach employs distinct detection modules and their fusion, each capitalizing on the strengths of different sensor modalities.

• **Height map:** As a basic obstacle detector, we employ a height map generated from the LiDAR point cloud, similar to the one presented in [41]. This method analyzes height differences in a grid, identifying obstacles exceeding a threshold. Grid cells with no such differences are deemed part of a plane. We use a polar grid map with 20 cm as the threshold. This detection mechanism serves as a backup for emergency situations. The first branch in Figure 4 depicts a height map-based perception process, where points are assigned colors based on their height to create a visual representation of the surrounding environment’s topography then grid cells within that contain points exceeding the predetermined vertical distance threshold are identified. Finally, points situated within these designated grid cells are marked in red to clearly indicate potential obstacles. Our obstacle detector using a height map in a polar grid is open source and available online (Available at: https://github.com/luis2r/Polar-Height-Map (accessed on 5 February 2024)).

• **Instance segmentation:** For object detection, we employ Mask R-CNN [42], a powerful instance segmentation algorithm that extracts coordinates of bounding boxes and masks for each object instance detected in the image. Our system categorizes these objects into eight categories: car (including vans, trucks, and buses), bicycle (including motorcycles), pedestrian, red traffic light, yellow traffic light, green traffic light, stop, and emergency vehicle. Code is open source and available online (Available at: https://github.com/luis2r/Instance-segmentatio-CARLA (accessed on 22 January 2024)).

• **Disparity/Depth estimation:** The third branch demonstrates how disparity estimation leads to the generation of either an RGB-D image or an RGB point cloud. This is achieved by processing a pair of stereo images. This module allows for the use of any disparity algorithm. In our case, we leverage the efficient large-scale stereo matching (ELAS) algorithm, commonly employed in autonomous driving applications [43].

One of the key advantages of ELAS is its computational efficiency. ELAS is known for its ability to efficiently compute dense stereo correspondences, making it suitable for real-time applications such as ours. This efficiency allows our system to process stereo image pairs rapidly (approximately 5 hz for our setup), enabling the timely generation of 3D point clouds for perception tasks.

Moreover, ELAS exhibits robustness in handling various scene complexities and texture variations. It is capable of producing accurate depth estimates even in challenging environments with occlusions, textureless regions, and varying lighting conditions. This robustness ensures reliable perception capabilities across a wide range of scenarios encountered in autonomous driving environments.

However, the modular nature of our pipeline allows for the incorporation of other typical disparity estimation algorithms, such as semi-global matching (SGM) [44] or other modern stereo matching algorithms based on deep learning such as IGEV-Stereo [45].

• **Fusion with stereo camera:** Since the image used for instance segmentation is also the left image of the stereo camera, we leverage this for 3D detection and classification. The bounding box coordinates and pixel mask of each object instance detected in 2D are mapped to corresponding 3D points in the organized point cloud. This 3D point cloud inherits the color and the image’s row and column structure but expands it with 3D/depth information. This fusion process creates an RGBD point cloud with color and 3D/depth information for each object instance, enabling accurate classification and positioning.

The third branch, Figure 4, illustrates an example of a point cloud constructed from the stereo camera, while the fourth branch shows an example of instance segmentation corresponding to the same scenario. Finally, the fusion of these two branches enables the detected objects to be mapped onto the RGBD point cloud, providing a comprehensive visualization of their positions within the 3D environment.

While this method can detect both static and dynamic objects, it is not the primary system detector in our architecture due to its non-real-time operation with a maximum of 5 frames per second. Nevertheless, the detailed information it provides on traffic light states, unavailable from LiDAR, justifies its inclusion despite the latency. Dynamic object detection (cars, bicycles, and pedestrians) in this fusion serves as a backup for emergencies, and we currently do not track these objects.

• **Three-dimension detection in point clouds (dynamic objects):** For 3D detection in LiDAR point clouds, we leverage the PointPillars algorithm [46]. This algorithm provides regression of 3D bounding boxes of objects and their orientation related to the ego vehicle. PointPillars demonstrates commendable performance in real-world autonomous driving scenarios. Its pillar representation retains valuable spatial information while maintaining computational efficiency, a critical factor for real-time applications. Moreover, it effectively leverages the strengths of LiDAR data, including its ability to handle occlusions and perform reliably in diverse lighting conditions.

• **Tracking:** The perception stack (in point clouds) detailed in this section, which supplies inputs to the risk assessment module responsible for determining finite state machine (FSM) graph states, consists of two integral components: (i) **pose estimation** (bounding boxes) and (ii) **multi-object tracking** (MOT) modules. Having a stable and precise state estimation, both for the ego-vehicle and dynamic objects in the surroundings, is important to transition across the states.

Regarding the **multi-object tracking** module, we based it on the one proposed by [47]. This approach, known as *simple online and realtime tracking* (SORT), divides the tracking task into three sub-tasks: detection, data association, and state estimation.

As detailed earlier, objects are detected in the LiDAR frame using PointPillars. This detection model differs from the ones adopted originally by the SORT tracker. The detected objects’ bounding boxes are projected into the world frame using the ego-vehicle estimated pose, and their respective poses are matched in the data association step. At this point, we keep the SORT tracker data association in the 2D space, in which we compute the intersection over union (IoU) of the top-down projection of the bounding boxes. Our assumption is that two dynamic objects do not overlap in the X–Y plane.

As for the state estimation, we use a Kalman Filter-based approach, in which our goal is to estimate the 3D position of the bounding box center (x,y,z), 3D dimensions of the bounding box $\left(s_{x},s_{y},s_{z}\right)$, the yaw angle ψ, and the 3D velocities $\left(\dot{x},\dot{y},\dot{z}\right)$ of the tracked object in the International System of Units and in the global reference frame. This space state differs from the SORT paper state, in which the estimated state is performed in the pixel space. Also, notice that the vehicle pose estimation is important in this step, as we are tracking in the global reference frame.

As we represent this state vector as $S_{obj}={\left(x,y,z,s_{x},s_{y},s_{z},\psi ,\dot{x},\dot{y},\dot{z}\right)}^{T}$, we need to define both the state propagation and observation model matrices.

The state propagation we adopted assumes constant linear velocity between detections so that we can propagate the position using the propagation matrix *F* from Equation (Equation 1).
(1)$$F=\left[\begin{array}{ccc} I_{3x3} & 0_{3x4} & \Delta T\cdot I_{3x3}\\ 0_{4x3} & I_{4x4} & 0_{4x3}\\ 0_{3x3} & 0_{3x4} & I_{3x3} \end{array}\right],$$
where *I* represents the identity matrix, 0 the zero-filled matrix, and ΔT represents the period between predictions. The subscripts indicated with MxN represent the matrix number of rows *M* and columns *N*, respectively.

Regarding the observation model matrix, since we directly obtain the 3D position, dimensions, and yaw angle from our detection module, our observation matrix is defined by $H=I_{7x10}$.

The third branch in Figure 4 illustrates detection and tracking in 3D, with LiDAR serving as input to the 3D detector. In our case, the 3D detections of point pillars are fed into the SORT algorithm, which we have modified for tracking. Ultimately, for visualization, each tracked object is assigned a distinct color.

• **Prediction:** Our system employs a prediction-based approach to ensure safe navigation by anticipating the movements of surrounding objects. This approach utilizes a simple motion model based on the object’s current speed, as estimated by the tracking system and this model assumes constant velocity for each object, providing a first-order approximation of their trajectories.

The prediction formula for a linear motion model is defined by Equation (Equation 2).
(2)$$x\left(\Delta t\right)=x_{o}+v\Delta t$$
where $x_{o}$ is the current pose, *v* is the actual velocity of the surrounding object, and Δt represents the time interval for prediction (5 s). Finally, $x\left(\Delta t\right)$ is the predicted pose in the interval Δt.

While more complex prediction models exist, opting for a simple linear model ensures computational efficiency. This linear motion model for future pose prediction demands fewer computational resources, making it suitable for real-time applications in autonomous driving.

### 3.3. Risk Assessment

Our autonomous driving system employs a dedicated collision risk assessment (CRA) module to continuously evaluate potential threats posed by both dynamic (cars, pedestrians, bicycles) and static surrounding objects. This module integrates the current and future positions of surrounding objects, obtained from the previous stage, with the ego vehicle’s planned path for risk assessment.

• **Zoned Risk Evaluation:** The planned path is divided into two zones, each reflecting different risk levels based on distance from the ego vehicle: a High Risk Zone (0–4 m) and a Moderate Risk Zone (4–40 m). For risk evaluation, the path ahead is divided into two zones, each carrying different risk levels based on Euclidean distance. These zones are corridors created from the waypoints of the planned path (essentially buffer zones extending 40 m ahead of the ego vehicle). The width of these corridors matches the width of the ego car.

Any object (static or dynamic) whose current position intersects either zone is considered a potential collision threat. The intersection point of predicted trajectories with the ego vehicle’s path is also considered. We assume that our lateral model predictive control (MPC) controller guarantees that the ego car will pass exactly through these corridors.

The identified potential points of collision and object information, including type, distance, and predicted trajectory, are reported to the decision-making module for determining appropriate speed adaptations. Figure 5 visually illustrates the risk assessment process, showcasing three points as examples. Note that other surrounding objects and their predicted trajectories are currently ignored unless they enter the relevant risk zones or directly influence the planned path.

### 3.4. Decision Making

The decision-making module utilizes a synchronous Moore finite state machine (FSM) to orchestrate actions based on inputs from the collision risk assessment (CRA) module. The FSM employs a binary encoding scheme for inputs as shown in Table 2.

The finite state machine (FSM) consists of four key states. In this FSM, the next state is determined solely by the current input, not the previous state. Each state governs specific speed control behaviors.

• **Drive State (S1):** No obstacles impede the vehicle’s progress. Target speed is set to a maximum of 8.8 m/s (31.68 km/h). This speed was empirically chosen to prevent penalties from the MinSI metric on Leaderboard 2, which considers the average speed of agents in the simulation. While Leaderboard 1 has permissive speed requirements, Leaderboard 2 introduces the MinSI metric, requiring a speed that balances collision avoidance and adherence to the minimum speed limit.

• **Follow the Leader State (S2):** The CRA reports an obstacle (static or dynamic) ahead of the ego vehicle, triggering dynamic speed adjustments. Speed is adjusted based on distance and time to collision (TTC), calculated using the ego vehicle’s current speed and distance to the obstacle.

• **Red Light State (S3) and Stop Sign State (S4):** These states mirror the “Follow the Leader” logic, utilizing TTC to achieve controlled stops at designated locations. The vehicle decelerates smoothly, ensuring compliance with traffic rules and safety.

Figure 6 depicts the state transition diagram, visually representing the FSM’s logic. Table 3 provides a detailed state transition table. The decision-making module employs a straightforward yet effective FSM structure for robust decision making. Speed control strategies adapt dynamically to varying conditions, ensuring safe and efficient navigation. The module seamlessly integrates with other components of the autonomous driving system, including perception and control modules. The FSM operates synchronously at 10 Hz, aligning with sensory data capture rates.

The current approach uses the actual velocity of the ego vehicle to calculate TTC, after which velocity adjustments are made to prevent collisions. This method is simple, fast, suitable for quick estimations, easy to implement, and computationally efficient. However, it assumes constant velocity and ignores potential future trajectory changes. The TTC formula is: TTC=Distance/RelativeVelocity.

### 3.5. Control

The *control layer* generates steering, throttle, and brake commands to keep the agent on the planned trajectory. This goal is achieved through two closed-loop control mechanisms that receive desired vehicle trajectory information from the navigation layer’s decision-making and local path-planning modules. These modules set the desired trajectory and velocity into the agent’s action space. The closed-loop controls translate reference values into actual control actions for braking, throttle, and steering, which are then sent directly to the simulator for execution.

For longitudinal control, the decision-making module (FSM) calculates the desired agent’s velocity, which is used to compute the final velocity. A proportional–integral–derivative (PID) controller then ensures the agent follows this desired reference.

#### Lateral Control (MPC)

Lateral control employs model-based predictive control (MPC) to generate the steering signal. MPC is a control strategy that relies on mathematical models of systems to predict future behavior and compute optimal control actions over a finite time horizon *H*. It formulates an optimization problem to minimize a predefined cost function, which typically accounts for control objectives and constraints on system variables. MPC computes a sequence of control actions, one for each time step (Δt), that optimize the cost function over the prediction horizon and applies only the first control action to the system. This process repeats at each time step, with the prediction horizon shifting forward in time, allowing MPC to continually adjust control actions based on updated measurements and changing system conditions.

The limitations defining the vehicle’s motion model are inherently non-holonomic. Non-holonomic systems pose unique challenges in control and navigation due to their inherent constraints on motion that limit the degrees of freedom. Unlike holonomic systems, which can move freely in any direction, non-holonomic systems, such as car-like robots or vehicles, are restricted in their movements. These constraints often manifest as limitations on the system’s velocity, acceleration, or steering angles, making it difficult to achieve desired trajectories or execute complex maneuvers efficiently [28]. Thus, planning feasible paths while adhering to the non-holonomic constraints adds complexity, requiring advanced algorithms and optimization techniques.

Given the assumption of wheels rolling without slipping, only the kinematic equations are pertinent, and the lateral dynamic effects can be disregarded [48]. Hence, the considerations outlined thus far yield the following kinematic model [49]:(3)$$\left[\begin{array}{c} \dot{x}\\ \dot{y}\\ \dot{\theta }\\ \dot{\kappa } \end{array}\right]=\left[\begin{array}{c} \cos\theta \\ \sin\theta \\ \kappa \\ 0 \end{array}\right]v+\left[\begin{array}{c} 0\\ 0\\ 0\\ 1 \end{array}\right]\tau ,$$
where $\tau =\dot{\phi }/\left(D_{bl}\cos^{2}\phi \right)$.

Incorporating the motion constraints into the optimization problem involves introducing the third power of Δt based on Equation (Equation 3) [50], where *v* is determined by the decision-making module (treated as constant in the optimization). The cost function is formulated as the summation of quadratic differences between the decision variables and the reference path,
(4)$$L_{ref}=C_{x}\frac{1}{2}{\left(x-x_{ref}\right)}^{2}+C_{y}\frac{1}{2}{\left(y-y_{ref}\right)}^{2}+C_{\theta }\frac{1}{2}{\left(\theta -\theta_{ref}\right)}^{2}+C_{\kappa }\frac{1}{2}{\left(\kappa -\kappa_{ref}\right)}^{2},$$
and also, the quadratic of τ
(5)$$L_{\tau }=C_{\tau }\frac{1}{2}{\left(\tau \right)}^{2},$$
where $C_{x}$, $C_{y}$, $C_{\theta }$, $C_{\kappa }$, and $C_{\tau }$ are cost weights manually tuned. The chosen parameters are shown in Table 4.

### 3.6. Localization

In order to perform the ego-vehicle **pose estimation**, we fused the relative transforms obtained using an odometry source (in our case, visual-inertial odometry—VIO), the inertial measurement unit (IMU) orientation, and the global navigation satellite system (GNSS) position using an extended Kalman filter (EKF) approach, as in Figure 7. The inputs of our stack are the camera image, the IMU orientation, the GNSS coordinates, and the sensor calibration (external reference frames’ relative transformation). The output elements of the pose estimation stack are estimated pose and its uncertainty.

The VIO estimation is responsible for estimating $T_{cam_{t}}^{cam_{t-1}}$, which represents the pose transformation matrix of the current camera frame with respect to the previous, and the estimation uncertainty covariance matrix, $\Sigma_{vio_{t}}$.

While the GNSS is responsible for providing the global geographic coordinates, the IMU provides the linear acceleration, angular velocity, and 3D orientation at a higher frequency. We then synchronize both the 3D orientation and global coordinates in order to provide $T_{imu_{t}}^{W}$, which represents the transformation matrix of the IMU frame relative to the world frame, and its uncertainty, $\Sigma_{global_{t}}$. In our case, the IMU and GNSS are represented by the same reference frame, but we left them illustrated in the diagram for the sake of clarity. Also, the geographic coordinates provided are then converted to a plane projection coordinate system.

The input poses, $T_{cam_{t}}^{cam_{t-1}}$ and $T_{imu_{t}}^{W}$, and the sensor calibrated, are then provided to the EKF and then converted to a common reference frame internally. The goal is to estimate the 6DoF pose of the agent frame with respect to the world frame, $S_{agent}={\left(x,y,z,q_{x},q_{y},q_{z},q_{w}\right)}^{T}$, where: (x,y,z) are the global coordinates, easting, northing, and altitude, respectively, and $\left(q_{x},q_{y},q_{z},q_{w}\right)$ represents the four components of the quaternion that represents our agent’s orientation. For each relative pose received, $T_{cam_{t}}^{cam_{t-1}}$, the EKF performs a system prediction, which implies accumulating drift until a global pose, $T_{imu_{t}}^{W}$, is received and the state update is performed.

We emphasize that this pose estimation module is also modular, so that the back end (in this case, the EKF), the methods used for estimating the relative transforms, and the source of the global pose estimation do not need to be the same as the ones we used in this project.

In practice, for estimating the relative pose transformations using VIO, we used the RTabMap ROS implementation (Available at: https://github.com/introlab/rtabmap_ros (accessed on 5 February 2024)). RTabMap is known for its estimation robustness, and its full functionalities are widely used in SLAM applications. As for the EKF implementation, we used the GTSAM implementation (Available at: https://gtsam.org/doxygen/4.0.0/a03631.html (accessed on 5 February 2024)). While GTSAM is known for implementing solutions using factor graphs, it also implements a very convenient interface for representing pose transformations and implements the 3D pose extended Kalman filter off-the-shelf. Finally, our localization stack is open source and available online (Available at: https://github.com/cabraile/LRM-Localization-Stack-2023 (accessed on 5 February 2024)).

## 4. Hybrid Architecture for Mapless Autonomous Driving

This section introduces our hybrid architecture for mapless autonomous driving, designed to navigate challenging scenarios like the CARLA Leaderboard’s SENSORS track. Building upon our modular pipeline described in previous sections, we leverage robust obstacle detection, risk assessment, and decision-making modules while replacing traditional map-based planning with an end-to-end path planner named the CNN-planner [10].

Conventional map-based approaches often struggle in dynamic environments lacking accurate maps. We overcome this limitation by using an improved version of the CNN-planner for mapless situations. This planner generates a set of waypoints and utilizes sensor fusion in the BEV space as its primary input.

This fusion is named **BEVSFusion,** which is a rich data structure that seamlessly fuses high-level commands and point clouds.The BEVSFusion structure receives data from three sources that are processed in the following way.
**Stereo Camera**: We utilize a pair of cameras with a specific field of view, resolution, and baseline. Disparity maps are calculated with the ELAS algorithm and projected into a point cloud. The stereo point cloud is transformed from the camera coordinate system to the BEV coordinate system using the transformation matrix $T_{cam}^{BEV}$.**LiDAR**: The LiDAR point cloud is directly transformed to the BEV coordinate system using a $T_{LiDAR}^{BEV}$ transformation matrix. LiDAR points are then rasterized in an RGB image where a colormap encodes height information (blue for ground, yellow for above sensor). Empty pixels are filled with black.**High-Level Commands**: Global plan commands are converted from the world frame to the BEV frame using $T_{W}^{BEV}$ and rasterized as colored dots in BEV space (blue for turn right, red for turn left, white for straight, and green for lane follow). Additionally, we rasterize a straight line connecting two adjacent high-level commands. This connection enhances the representation of the order and sequence of points within the raster, facilitating interpretation and providing additional information to the CNN network.

Finally, these three processed elements (rasterized LiDAR, stereo, and high-level commands) are stacked into a single 9-channel image (BEVSFusion). This unified structure integrates spatial, depth, height, color, and high-level command information for robust path planning.

BEVSFusion serves as the input to CNN of the path planner. The CNN-Planner can be represented as a function:(6)w=CNN-Planner(BEVSFusion),

This CNN is an architecture for the regression of a sequence *w* of dense waypoints for the ego vehicle’s trajectory. Each waypoint in the sequence $w=\{w_{1},\ldots ,w_{n}\}$ represents a point with (x,y) coordinates in the ego-car coordinate system. $w_{1}$ corresponds to the closest point to the ego car, while $w_{n}$ denotes the furthest point on the planned trajectory. *w* is transformed from the ego coordinate system to the world coordinate system using the $T_{ego}^{W}$ transformation matrix.

Compared to the original CNN-planner implementation, which generates an output of 50 *w* waypoints, our approach yields 200 *w* waypoints. These waypoints cover a distance of 40 m, resulting in a longer path. This extended path not only provides a larger area for monitoring potential issues but also ensures more precise guidance for the vehicle.

Figure 8 visually illustrates the process, highlighting the creation of BEVSFusion through sensor fusion and its integration as input for the CNN-planner and generating the planned trajectory. Finally, the path *w* in the world coordinate system is followed using an MPC controller.

For our CNN-Planner, we train a ResNet 18 over 100 epochs using the Adam optimizer [51] and the MSE error (Equation (Equation 7)).
(7)$$L=MSE\left(y,\hat{y}\right)=\frac{\sum_{i=0}^{N-1}{\left(y_{i}-{\hat{y}}_{i}\right)}^{2}}{N}$$
where *y* is the predicted path and $\hat{y}$ is the ground truth path.

## 5. Experiments and Results

### 5.1. Experimental Setup

Our research adopts the robot operating system (ROS) as the unifying framework for both our modular and hybrid driving architectures. ROS’s publisher–subscriber communication paradigm [52] facilitates efficient data exchange between components, enabling a flexible and scalable system design. The ROS master node indexes and coordinates components, while peer-to-peer messaging enables direct communication between nodes [36]. This structure streamlines the development and integration of multi-component systems, particularly in applications like autonomous driving and robotics.

Our autonomous driving agents implement all modules described, for the perception layer we have the following.
Two monocular cameras with 71° field of view (FOV) each are combined to form a stereo camera for 3D perception, producing a pair of rectified images with dimensions of 1200×1200 pixels. The baseline of our stereo camera is 0.24 m. We utilize the ELAS algorithm [43] to generate 3D point clouds from the stereo images.LiDAR sensor: 64 channels, 45° vertical field of view, 180° horizontal field of view, 50 m range. Our system utilizes a simulated LiDAR collecting around one million data points per scan across 64 vertical layers.LiDAR and stereo cameras are centered in the x-y plane of the ego car and mounted at 1.8 m height.GNSS and IMU: For localization and ego-motion estimation.CANBus: Provides vehicle internal state information such as speed and steering angle.

Our modular architecture additionally utilizes an OpenDrive map pseudo-sensor for route planning and an ObjectFinder pseudo-sensor, used exclusively for dataset creation, provides ground-truth information about dynamic and static objects within the CARLA simulator.

### 5.2. Time Execution Results

This section provides a comprehensive analysis of the execution time for each module within the modular pipeline. Our tests were conducted locally on a high-performance computer equipped with a 16-core Intel Core i9-9900KS processor, 64 GB of RAM, and two Nvidia RTX GeForce 2080Ti graphics cards.

Table 5 presents the time taken by each module to perform its respective tasks, offering insights into the efficiency of our approach. Tasks requiring only CPU show very short execution times, all below 1 ms, except for the disparity/depth module (210 ms). While not ideal, this module is not our primary option for obstacle detection, as it is only used for traffic lights and long-range pedestrian detection.

Among the GPU modules, instance detection is the most time-consuming, taking 198 ms. Its output is merged with the disparity/depth module’s output (as described in Section 3.2) for 3D detection of traffic lights and long-range people. Our core obstacle detection module, the 3D PointPillars detector, runs at an average of 95 ms, which is a processing rate close to 10 Hz, similar to the frequency of current LiDAR sensors. This allows for real-time processing. Finally, the CNN-planner module, leveraging a lightweight ResNet-18 for path generation in each frame, is the fastest module executing in GPU, requiring only 5 ms. Moreover, these results fulfill the real-time execution demands of our agents, while also highlighting areas for potential optimization to further improve overall system performance.

### 5.3. Metrics

Autonomous vehicles are heterogeneous and complex systems, orchestrating sensing, perception, decision making, planning, control, and health management. Evaluating the performance of these complex systems requires a holistic approach, going beyond individual evaluation to assess the harmony of the entire system.

Traditionally, unit tests analyze individual components, seeking malfunctions and quantifying their performance with metrics like accuracy, recall, and precision (e.g., for classification algorithms [53,54]). Integration tests take a broader perspective, examining the interplay between two or more components (e.g., obstacle detection and avoidance). Finally, system tests encompass the entire system, evaluating the harmonious collaboration of all its components [55,56]. However, a standardized methodology for comprehensively assessing and comparing the complete performance of autonomous driving systems remains a challenge. The CARLA Leaderboards offer a standardized benchmark for evaluating autonomous driving systems, providing diverse sensor configurations and software architectures.

These leaderboards immerse the autonomous system, or "agent", in simulated urban environments. Each scenario throws diverse challenges, varying in cityscapes, traffic areas (highways, urban roads, residential areas, roundabouts, unmarked intersections), route lengths, traffic density, and weather conditions. Moreover, each route incorporates traffic situations inspired by the NHTSA’s (National Highway Traffic Safety Administration of the United States) pre-crash typology [57], encompassing diverse scenarios like:Control loss without prior action.Obstacle avoidance for unexpected obstacles.Negotiation at roundabouts and unmarked intersections.Following the lead vehicle’s sudden braking.Crossing intersections with a traffic-light-disobeying vehicle.

Leaderboard 2 expands this scenario, adding:Lane changes to avoid obstacle blocking lanes.Yielding to emergency vehicles.Door obstacles (e.g., opened car door).Avoiding vehicles invading lanes on bends.Maneuvering parking cut-ins and exits.

To evaluate agent performance in each simulated scenario, CARLA Leaderboards employ a set of quantitative metrics that capture not only route completion but also adherence to traffic rules and safe driving practices. This metric assesses the entire system’s performance, transcending mere point-to-destination navigation. It factors in traffic rules, passenger and pedestrian safety, and the ability to handle both common and unexpected situations (e.g., occluded obstacles and vehicle control loss).

Key Metrics: **Driving Score (DS)**: The main metric of the leaderboards, calculated as the product of route completion percentage (Ri) and the infraction penalty (Pi) of the *i*-*th* route, (RiPi). This metric rewards both efficient navigation and adherence to safety regulations. **Route Completion (RC)**: Percentage of the route distance successfully completed by the agent of the *i*-*th* route, (Ri). **Infraction Penalty (IP)**: $\left(\prod_{j}^{ped,veh,\ldots ,stop}{\left(p_{j}^{i}\right)}^{\#infractions_{j}}\right)$. Aggregates all types of infractions triggered by the agent as a geometric series. Each infraction reduces the agent’s score, starting from an ideal base of 1.0. Specific infraction types and their penalty coefficients include:Collisions with pedestrians (CP)—0.50.Collisions with other vehicles (CV)—0.60.Collision layout (CL)—0.65.Running a red light (RLI)—0.70.Stop sign infraction (SSI)—0.80.Off-road infraction (ORI)—percentage of the route will not be considered.

Additional Leaderboard 2 Metrics:Scenario timeout (ST)—0.70.Failure to maintain minimum speed (MinSI)—0.70.Failure to yield to emergency vehicle (YEI)—0.70.

Under certain circumstances, the simulation will be automatically terminated, preventing the agent from further progress on the current route. These events include:Route deviations (RD);Route timeouts (RT);Agent blocked (AB).

After all routes are completed, global metrics are calculated as the average of individual route metrics. The global driving score remains the primary metric for ranking agents against competitors. By employing comprehensive evaluation frameworks like CARLA Leaderboards, researchers and developers can gain valuable insights into the strengths and weaknesses of their autonomous driving systems, ultimately paving the way for safer and more robust vehicles that perform harmoniously as a whole, not just as a collection of individual components. For further details on the evaluation and metrics, visit the leaderboard website (Available at: https://leaderboard.carla.org/#evaluation-and-metrics (accessed on 5 February 2024)).

To evaluate an agent’s performance, it must be submitted to the online evaluator (Available at: https://eval.ai/web/challenges/challenge-page/2098/overview (accessed on 5 February 2024)). The specific routes and the cities used are secret and confidential. For Leaderboard 1, 10 routes are chosen and each is evaluated 10 times under varying lighting and weather conditions. Each route is roughly 1 km long, meaning an agent completing all routes at 100% would cover approximately 100 km in total. Leaderboard 2 features increased difficulty compared to Leaderboard 1, with routes 10 times longer and presenting more complex scenarios. Agents must navigate these scenarios, including overtaking obstacles or yielding to emergency vehicles.

### 5.4. Datasets

To train the diverse components of our autonomous driving agents, we generated three comprehensive datasets. These datasets were created using CARLA simulator version 0.9.13 under a range of lighting and weather conditions (day, night, rain, fog) and across distinct urban environments in the CARLA towns: Town01, Town3, Town4, Town06, and Town12. These environments encompass downtown areas, residential neighborhoods, rural landscapes, and diverse vegetation.
**Instance Segmentation Dataset**: We constructed a dataset of 20,000 RGB images with variable resolutions ranging from 800×800 to 1400×1400 pixels. These images encompass seven object classes: car, bicycle, pedestrian, red traffic light, yellow traffic light, green traffic light, and stop sign.For labeling, we employed a semi-automatic approach for cars, bicycles, pedestrians, and stop signs, leveraging sensor instances provided by the CARLA simulator. Traffic lights and stencil stop signs, however, required manual annotation for greater accuracy. All annotations were stored in the COCO format. Finally, we trained a Mask-RCNN model implemented in mmdetection (Available at: https://github.com/open-mmlab/mmdetection (accessed on 5 February 2024)) for object detection and segmentation. Figure 9 showcases examples of detections achieved with our trained model. Our Instance Segmentation Dataset is available online (Available at: https://github.com/luis2r/Instance-segmentatio-CARLA (accessed on 22 January 2024)).**Three-Dimensional Object Detection Dataset:** This dataset comprises 5000 point clouds annotated with pose (relative to the ego car), height, length, width, and orientation for all cars, bicycles, and pedestrians. We leveraged the privileged sensor objects within the simulator to perform this automatic annotation. The data were subsequently saved in the KITTI format for compatibility with popular object detection algorithms. Using this dataset, we trained a PointPillars model adapted for our specific needs, implemented in the mmdetection3d framework (Available at: https://github.com/open-mmlab/mmdetection3d (accessed on 22 January 2024)).**Path Planner Training Dataset:** To train the path planner, we leveraged a privileged agent and the previously described sensors to collect approximately 300,000 frames. This agent granted access to ground-truth path information and provided error-free GPS and IMU data facilitating precise navigation. The point clouds from the LiDAR and stereo cameras were then projected and rasterized into 700 × 700 RGB images in the bird’s-eye view space. High-level commands like “left”, “right”, “straight”, and “lane follow” were transformed to the ego coordinate system using the command pose, then rasterized within the bird’s-eye image in the same way as pointclouds but with color-coded points for commands (red for left, blue for right, white for straight, and green for lane follow). The ground-truth road path consisted of 200 waypoints spaced 20 cm apart, originating at the center of the ego car.To simulate potential navigation errors and enhance error recovery learning, we introduced Gaussian noise to the steering wheel inputs in 50% of the routes used for dataset collection.

### 5.5. Results on CARLA Leaderboards

This section presents the performance of our modular and hybrid CaRINA agent architectures on the CARLA Leaderboards, demonstrating their effectiveness in both map-based and mapless navigation tasks. To validate our models, we utilized the leaderboards provided by the CARLA team (Leaderboard 1 and Leaderboard 2). We employed the Track MAP from the two benchmarks to assess our modular architecture, and the Track SENSORS were used to evaluate our hybrid architecture, which does not require a map for navigation.

**Navigation using Map (Leaderboard 1 and 2 Track MAP):** We employed our modular CaRINA stack for map-based navigation using OpenDRIVE format as mentioned in previous sections.

Table 6 illustrates the results for Leaderboard 1 on the track MAP. On the Track MAP, we secured third place in the driving score metric (DS = 41.56) and the highest score in route completion (RC = 86.03) among all competitors using our modular pipeline. These achievements highlight the combined strength of our CaRINA modules.

Table 7 shows the evaluation in the track MAP of Leaderboard 2. We achieved second place in the driving score (DS = 1.14) and route completion (RC = 3.65%), only narrowly surpassed by another modular architecture. Importantly, our off-road infraction penalty in this track (ORI = 0.0) emphasizes the seamless navigation facilitated by the map. This compares favorably to all methods on both Leaderboard 1 and 2 Track MAP, where some map-based approaches based on TF++ [58] and MMFN [20] also achieve an ORI = 0.0.

**Table 6 sensors-24-02097-t006:** Results: CARLA Leaderboard 1, Track MAP.

Team	Method	DS	RC	IP	CP	CV	CL	RLI	SSI	ORI	RD	RT	AB
Anonymous	Map TF++	61.17	81.81	0.70	0.01	0.99	0.00	0.08	0.00	0.00	0.00	0.00	0.55
mmfn	MMFN+(TPlanner) ^1^	59.85	82.81	0.71	0.01	0.59	0.00	0.51	0.00	0.00	0.00	0.62	0.06
**LRM 2023**	**CaRINA agent**	41.56	**86.03**	0.52	0.08	**0.38**	0.13	1.6	0.03	0.00	0.04	0.05	1.29
RaphaeL	GRI-based DRL [59]	33.78	57.44	0.57	0.00	3.36	0.50	0.52	0.00	1.52	1.47	0.23	0.80
mmfn	MMFN [20]	22.80	47.22	0.63	0.09	0.67	0.05	1.07	0.00	0.45	0.00	0.00	1003.88
RobeSafe research group	Techs4AgeCar+ [60]	18.75	75.11	0.28	1.52	2.37	1.27	1.22	0.00	0.59	0.17	0.01	1.28
ERDOS	Pylot [61]	16.70	48.63	0.50	1.18	0.79	0.01	0.95	0.00	0.01	0.44	0.10	3.30
LRM 2019	CaRINA [10]	15.55	40.63	0.47	1.06	3.35	1.79	0.28	0.00	3.28	0.34	0.00	7.26

^1^ Available at: https://github.com/Kin-Zhang/mmfn (accessed on 2 January 2024).

**Mapless Navigation (Leaderboard 1 and 2 Track SENSORS):** We evaluated our hybrid CaRINA stack for mapless navigation.

Table 8 illustrates the results for leaderboard 1. In the Track SENSORS, our hybrid CaRINA agent achieved a route completion score of 85.01%, surpassing other autonomous driving methods primarily based on end-to-end learning. We also obtained a high driving score (DS = 35.36) compared to similar approaches (WOR [62], MaRLn [63], NEAT [64], AIM-MT [64], TransFuser [65], CNN-Planner [66], Learning by Cheating [67], CILRS [68], CaRINA 2019 [10]).

Table 9 presents the results for Leaderboard 2 on Track SENSORS, where our performance dominated the leaderboard with a driving score (DS) of 1.232 and a route completion (RC) of 9.55%, showcasing a significant difference, more than twice that of the second place’s performance. This highlights the effectiveness and competitiveness of our hybrid architecture for mapless navigation, surpassing the state-of-the-art end-to-end learning method zero-shot TF++ (variation of the TF++ method [58]).

Leaderboard 2 was used for the 2023 CARLA Challenge. We achieved first place in the Track SENSORS and second place in the Track MAP categories of the 2023 CARLA Challenge, with our modular and hybrid CaRINA agent versions, respectively. Two videos demonstrate the perception and navigation capabilities of our CaRINA agent on Leaderboard 2 (see videos 1 and 2 (available at: https://bit.ly/3PAyDfo (accessed on 5 February 2024), https://bit.ly/3Vx5Tbj) (accessed on 5 February 2024) for demonstrations).

### 5.6. Analysis and Discussion

The results in the previous section were obtained from the official CARLA Leaderboard. However, it is important to note that we only have access to the final scores for each metric, lacking additional details regarding the vehicle’s performance in individual traffic scenarios and their respective impacts on the overall score. In this section, we analyze the results based on both leaderboard scores and offline experiments conducted on a local machine. Through the insights gained from these offline experiments, we can draw conclusions about the performance of the modular and hybrid autonomous driving architecture.

#### 5.6.1. Modular Architecture

We assessed the modular navigation architecture on the MAP track in Leaderboards 1 and 2. In both cases, our RC (Route Completion) scores surpassed those of any other technique, indicating that our vehicles completed more trajectory segments than competing agents. Nevertheless, our agent incurred a lower IP (infraction penalty) than the top two agents with the highest DS (driving score). This outcome is primarily influenced by two types of infractions, namely RLI (running red light) and AB (agent blocked), as the remaining infraction metrics show no significant difference from the top two agents in Leaderboard 1.

In the first case (RLI), the complexity of the road network layout (especially the intersections) poses a significant challenge in correctly associating the traffic light with the vehicle’s current trajectory. Consequently, as the vehicle approaches the intersection, it may either pass the location where it should stop and wait for the traffic light or fail to detect it through its cameras. Certain road geometries and traffic light configurations in this scenario can cause the detector to struggle in identifying the traffic light’s status or associating it with the vehicle’s trajectory, especially when it deviates from the planned path.

During navigation and interaction at intersections, the agent repeatedly correlates the detected traffic lights with the planned path to consider only those signals directly affecting the agent’s mission. Addressing this issue would require enhancing the robustness of the intersection algorithm to variations and uncertainties. However, achieving generalization across numerous scenarios proves challenging. Another consequence is that delays in crossing intersections can result in penalties, potentially leading to collisions. Offline experiments revealed the particular difficulty the agent faced in this scenario. We adopted a conservative navigation strategy characterized by smooth speed adjustments and continuous monitoring of nearby obstacles, triggering emergency braking at the detection of imminent collisions based on the current vehicle planning and trajectory predictions of nearby obstacles. While successful on most routes, this approach has limitations, especially at busy intersections. Therefore, expediting vehicle decision-making and execution would be a potential solution to overcome current limitations.

In the second scenario, the vehicle struggles to pass an obstacle (e.g., object or another vehicle) that is stationary in its lane. Both situations concern the scene understanding within the perceptual system and decision making. Specifically, the second scenario presents an additional challenge when there is oncoming traffic in the opposite lane. In this case, in addition to recognizing the need for a lane change, the vehicle must also identify a gap in the traffic and promptly react to enter the gap. In offline experiments, we observed that due to the conservative driving style adopted, the vehicle is not always swift enough to enter a gap before the next approaching vehicle arrives.

Finally, it is worth noting that, despite having the highest RC among the agents on the leaderboard, the CV (collision with vehicles) is significantly lower than that of the other agents. This result emphasizes the effectiveness of the perception and decision-making systems in detecting and avoiding collisions with other vehicles.

#### 5.6.2. Hybrid Architecture

We assessed the hybrid architecture on the SENSORS track of Leaderboards 1 and 2. The “CaRINA hybrid” agent achieved significant route completion (RC), securing the sixth and first positions in Leaderboards 1 and 2, respectively. These outcomes are similar to those on the MAP track. However, in this track, the vehicle operates without access to map information, relying entirely on data-driven path planning. However, the considerable number of collisions with other vehicles (CV) significantly impacted the performance of the navigation architecture.

Based on offline experiments, we listed two scenarios that potentially affected the perception and navigation system, increasing the number of collisions with other vehicles. In the first scenario, lane changes were initiated due to potential obstructions in the current driving lane, such as other vehicles or objects. We observed that the data-driven path planning demonstrated superior adaptability, estimating lane-change trajectories in a broader range of scenarios compared to the modular navigation pipeline. However, the execution of these maneuvers resulted in more collisions with oncoming traffic in the opposite lane. The execution of lane change maneuvers also adopted a conservative approach with a gradual speed change profile and constant monitoring of collisions with nearby obstacles. Thus, the delay in completing the maneuver resulted in collisions in which other vehicles hit the agent’s side or rear, or the vehicle hit obstacles that approached while executing the maneuver. This behavior manifested in two additional metrics, apart from CV: AB (agent blocked), which is lower than the modular architecture due to the vehicle executing more lane-change and overtake maneuvers, and RD (route deviation), which occurs because the vehicle struggles to return to its trajectory after some collisions.

In the second scenario, the focus is on intersections, particularly when the vehicle fails to adhere to a red light signal. The vehicle approaches the intersection and attempts to cross it, but in most instances, it fails to avoid collisions with oncoming traffic. In certain situations, the vehicle stops midway through the intersection while trying to evade collisions and complete the maneuver. However, this behavior also results in the blockage of the vehicle (AB), contributing to intersection deadlock, or the vehicle running over road layouts (CL) and incurring off-road infractions (ORI).

#### 5.6.3. Comparison and Final Remarks

The results in the previous sections provided a comprehensive assessment of the performance of modular and hybrid architectures for autonomous navigation. The use of the CARLA simulator and Leaderboards 1 and 2 enabled a quantitative and qualitative evaluation of both approaches, providing valuable insights into their strengths and weaknesses. Accordingly, this section presents a brief overview of the results and observations related to both navigation strategies proposed in this paper.

The primary distinction between both approaches lies in their methodology. While the modular architecture relies on parsing the OpenDrive map to estimate trajectories and navigate, the hybrid approach employs a mapless data-driven path planning technique to guide the vehicle to its destination. Furthermore, the route completion (RC) of both approaches showed similarities across both leaderboards. This suggests the efficacy of the data-driven method in estimating trajectories in diverse urban scenarios, a notable challenge given the unfamiliarity of testing cities within the CARLA simulator. These cities feature different road network layouts and city landscapes. Additionally, the evaluation involved navigating under varying weather and light conditions, significantly impacting the performance of vision-based algorithms. The sensor fusion adopted in the data-driven approach, using images and point cloud, contributes to making the method more robust to adverse conditions, improving its adaptability and generalization.

Table 5 highlights a significant contrast between the modular and hybrid architectures, specifically in the frequency of trajectory planning. The modular approach generates trajectories using map information at a higher rate compared to the deep learning model in the hybrid architecture. Nonetheless, the frequency of the deep learning model is deemed adequate, considering the architecture’s time requirements during simulation. This divergence is caused by the simplicity of obtaining reference trajectories for navigation from a pre-existing map provided in the competition simulation. In contrast, the hybrid model employs sensory information to generate trajectories in an online and more reactive manner. This characteristic enabled the hybrid architecture to generalize effectively in scenarios demanding quick reactions, such as encountering obstacles in the vehicle’s path and addressing inconsistencies in map interpretation. This adaptability is evident in the AB (agent blocked) metric values on Leaderboards 1 and 2, where the hybrid vehicle (SENSORS track) outperformed the modular agent (MAP track). An additional noteworthy point is the performance variance in scenarios with obstacles on Leaderboard 2, which presents more complex scenarios than Leaderboard 1. In this context, the hybrid vehicle demonstrated superior performance compared to the modular agent, even covering a more extended route and encountering a broader range of scenarios.

Another important observation concerning the two approaches and the CARLA challenge is the complexity of the diverse driving scenarios. Apart from requiring effective perception, decision making, and planning systems, the challenge demands swift responses from the vehicle. For instance, when the vehicle needs to change lanes with traffic in the adjacent lane, it must identify a gap and react promptly. The offline experiments demonstrated various scenarios where the system’s components correctly identified these situations. However, the vehicle was not quick enough to execute maneuvers safely, resulting in collisions and other traffic infractions. We adopted a conservative driving style, which demands more time to react to scenarios involving interactions with other traffic participants. The smooth acceleration change curve led to dangerous situations, given that the behavior of other vehicles was designed to present complex and challenging scenarios for the autonomous agent. For example, in certain situations, due to lane changes or sudden brakes of the ego-vehicle (CaRINA agent), the surrounding vehicles collide with the rear of the ego-vehicle, as they were not designed to stop in such scenarios.

## 6. Conclusions

Our research not only proposes a versatile autonomous driving architecture, but also implements a robust approach to navigation. By blending the strengths of map-based and mapless paradigms within a unified framework. Integrating modularity with end-to-end path planning resulted in a holistic system that excels in both navigation styles. Modular simplicity facilitates transparent debugging and efficient issue identification, fostering continuous performance improvement. Our trajectory planning, despite using a minimalistic module compared to complex competitor models, competes impressively, exemplified by our route completion score in all tracks.

Training with a smaller dataset not only allows for focused problem debugging, but also fosters enhanced system interpretability through task-specific algorithms. This design permits a more efficient and effective development process. Ultimately, the success of the CaRINA agent, evident in its first-place in CADCH 2023 track SENSORS and second place in track MAP, testifies to the effectiveness and adaptability of our hybrid architecture.

### Challenges and Future Work

Despite the success of our models, we identified areas for improvement.

For traffic light detection, our strategy relies on the position of traffic lights to determine where to stop, but there may be configurations in unknown cities on the test server that we have not considered, leading to potential issues with our traffic light detection system. Our current reliance on traffic light position may not generalize to all scenarios. Exploring end-to-end or hybrid architectures for traffic light detection could address this limitation. Despite the success of our methods, we observe infractions, related to collision avoidance, especially in the area of collisions with vehicles. This is primarily due to lane changes requested by high-level commands (lane change left, lane change right), becoming hazardous when other high-speed vehicles are using the targeted lanes during lane changes. A potential solution could involve a new model and controller considering the other surrounding vehicle velocities or training an algorithm to adapt the speed during lane changes, particularly when other cars are traveling at high speeds.

It is important to acknowledge that a simple linear motion model has limitations. It may not accurately capture complex maneuvers or sudden changes in an object’s direction. Therefore, we acknowledge the need for exploring more sophisticated prediction models in future work, potentially incorporating acceleration data or historical movement patterns.

## Figures and Tables

**Figure 1 sensors-24-02097-f001:**
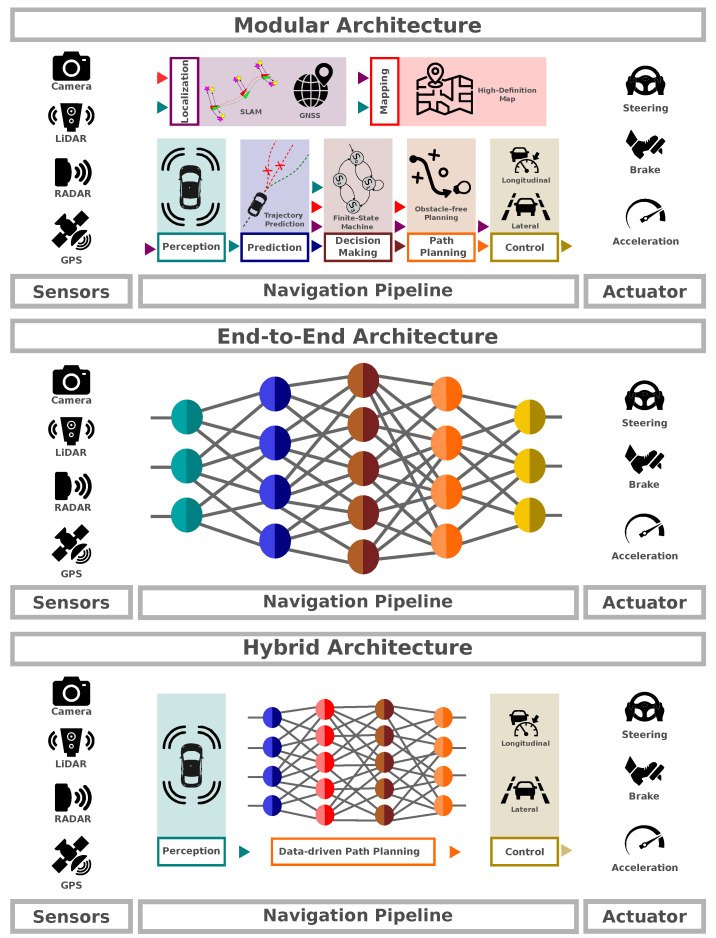
Example that illustrates the main differences among modular, end-to-end, and hybrid architectures.

**Figure 2 sensors-24-02097-f002:**
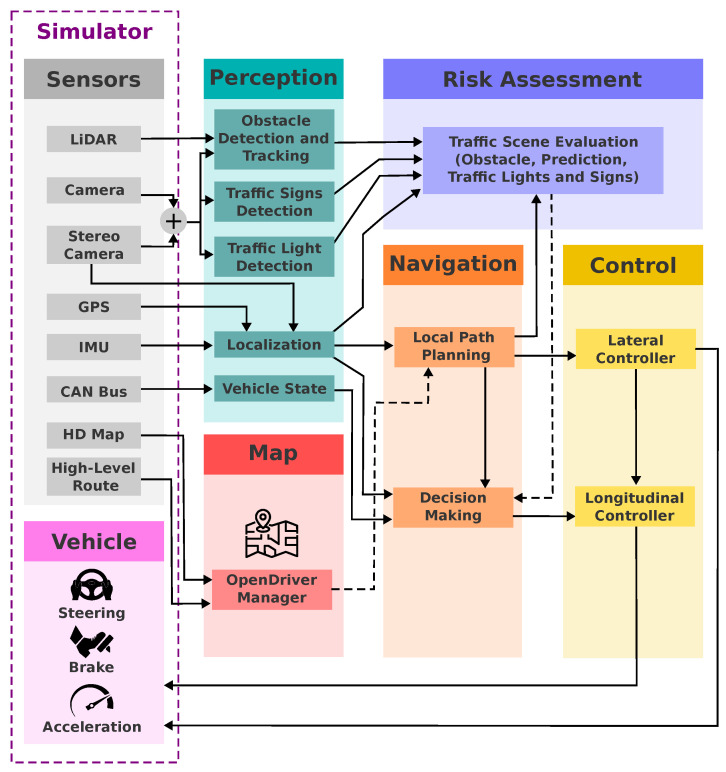
General design of the proposed modular architecture.

**Figure 3 sensors-24-02097-f003:**
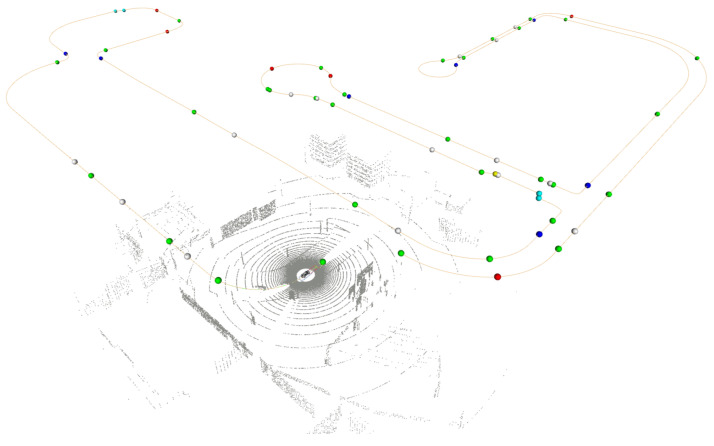
OpenDRIVE Map. The dots represent high-level commands with red (turn left), blue (turn right), green (keep lane), and white (go-straight).

**Figure 4 sensors-24-02097-f004:**
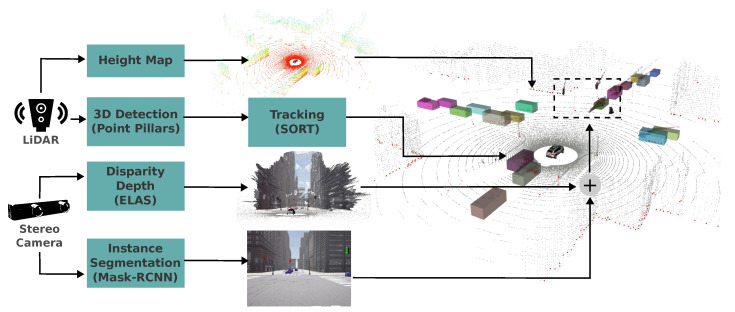
Perception module.

**Figure 5 sensors-24-02097-f005:**
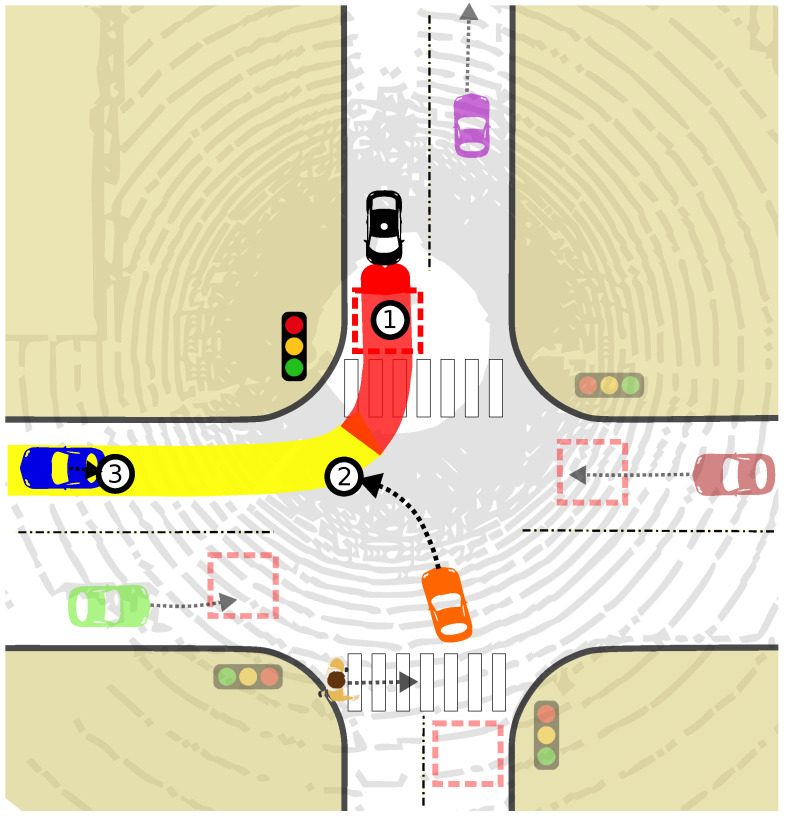
Risk assessment. *Point 1*—Zone of influence of a red traffic light. *Point 2*—The predicted trajectory of another car intersects the ego vehicle’s path in the yellow zone. *Point 3*—A parked car within the yellow zone is identified as a potential obstacle but receives lower priority compared to threats in the red zone.

**Figure 6 sensors-24-02097-f006:**
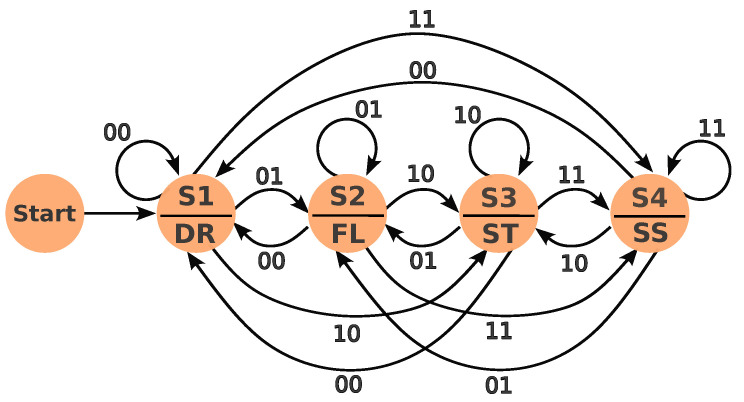
State transition diagram for the Moore finite state machine used in our decision-making module.

**Figure 7 sensors-24-02097-f007:**
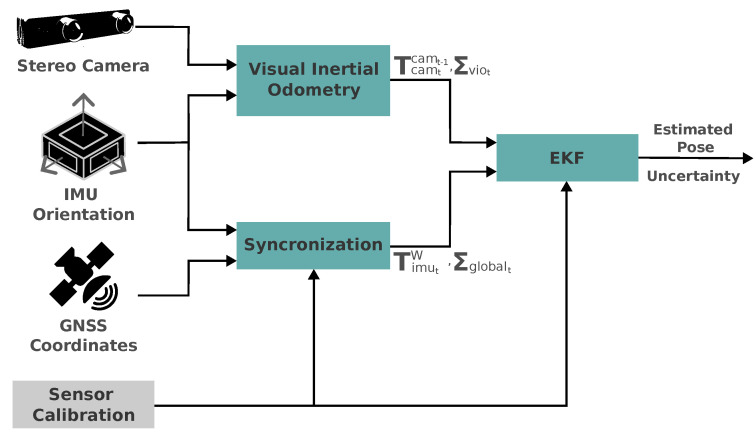
The pose estimation stack used in our perception module.

**Figure 8 sensors-24-02097-f008:**
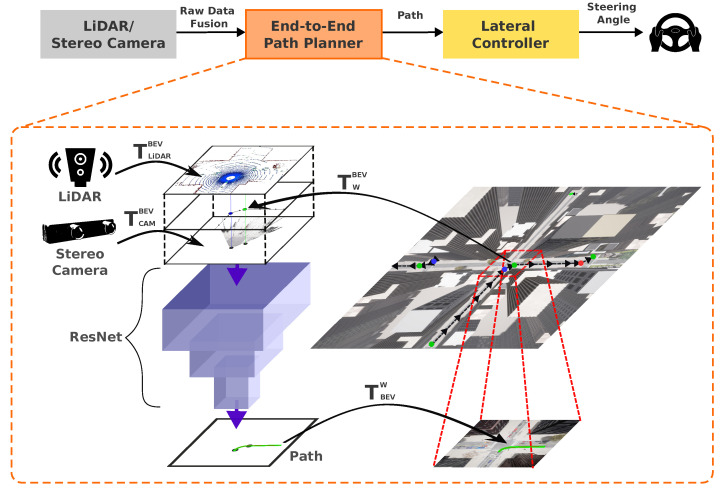
Our neural path planner takes as input BEVSFusion and the output is a list *w* (path) that is followed by the model predictive control (MPC) controller.

**Figure 9 sensors-24-02097-f009:**
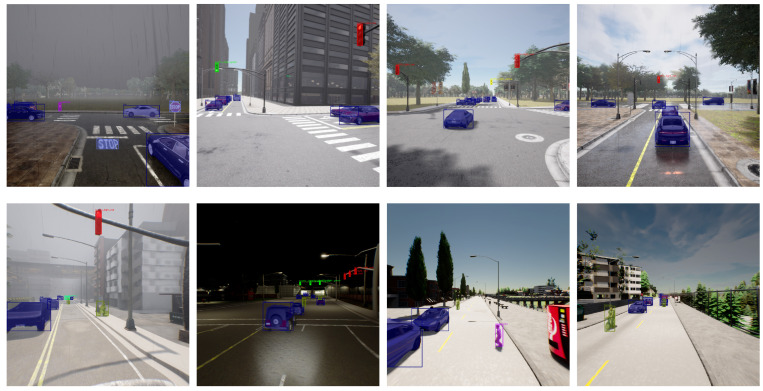
Our instance detection dataset includes annotations of eight classes: car, pedestrian, bicycle, stop sign, red traffic light, yellow traffic light, green traffic light, and emergency vehicle in different urban environments and weather conditions.

**Table 1 sensors-24-02097-t001:** Summary of related works.

Primary Study	Name	Type	Layers or Methods	Sensors or Inputs
Taş et al. [11]	BerthaOne	Modular	Sensing, Perception, Planning, Control, HMI, Communication	Radar, LiDAR, GNSS, IMU, Cameras, Stereo Camera, V2V, V2I
Fan et al. [12]	Apollo	Modular	Perception, Prediction, Planning, Control, HMI, Guardian, Localization, HD-Map, CANBus	Radar, LiDAR, GNSS IMU, Cameras, V2X, Untrasonic
Autoware [13]	Autoware	Modular	Sensing, Perception, Planning, Control, HMI, Localization, Map	Radar, LiDAR, GNSS, IMU, Cameras, Ultrasonic
Wei et al. [14]	CMU	Modular	Hardware, Perception, Mission Planning, Behavior Generation, Motion Planning	Radar, LiDAR, GNSS IMU, Cameras, V2V, V2I, Wheel speed sensor
Jo et al. [4]	A1	Modular	Sensor Interface, Autonomous Driving Algorithm, Actuator Interface, Development Interface	Lasers, GNSS IMU, Cameras
Shao et al. [15]	ReasonNet	End-to-End	ResNet (2D backbone), Transformer Encoder, PointPillars (3D backbone), GAT, CNN, MLP, GRU	Images (left, front, right and rear), LiDAR
Shao et al. [16]	InterFuser	End-to-End	ResNet (2D and 3D backbone), Transformer, GRU, MLP	Images (left, front, and right), LiDAR
Wu et al. [17]	TCP	End-to-End	ResNet (2D backbone), MLP, GRU	Image (front), Speed, High-level command, Goal
Casas, Sadat, and Urtasun [18]	MP3	End-to-End	CNN blocks (3D backbone), CNN (Map decoders), Probabilistic Reasoning	LiDAR, High-level command
Xiao et al. [19]	Multimodal CIL	End-to-End	CNN (2D backbone), MLP	Image (front), speed, Depth Image (front), High-level command
Zhang et al. [20]	MMFN	End-to-End	ResNet (2D and 3D backbone), VectorNet (3D backbone), GAT, MLP, GRU	LiDAR, Radar, Map, Goal
Cai et al. [21]	PMP-net	End-to-End	ResNet (2D and 3D backbones), Attention Mechanism, Gaussian Mixtute Model, MLP	Image (front), LiDAR, Radar, Map, Position, Goal, Speed
Vitelli et al. [22]	SafetyNet	Path Planning	PointNet (trajectory backbone) Transformer Encoder, MLP	Historical trajectory, Map, Goal
Song et al. [23]	IVGG LSTM	Path Planning	DCNN, CNN LSTM, MLP, VGG	Image (front, left and right rear view mirror)
Moraes et al. [24]	DeepPath	Path Planning	WideResNet38, DeepLabV3, MLP	Image (front), Position
Wang et al. [25]	CNN RawRNN	Path Planning	ResNet50 (2D backbone) LSTM, MLP	Image (front)
Hu et al. [26]	ST-P3	Path Planning	EfficientNet (2D backbone), DeepLabV3 (head), Attention Mechanism, GRU, MLP	Image (front-center, left, right; back-center, left, right), High-level command

HMI—human–machine interface; V2V—vehicle to vehicle; V2I—vehicle to infrastructure; V2X—vehicle to anything; CNN—convolutional neural network; MLP—multi-layer perceptron; GRU—gated recurrent unit; GAT—graph attention network; DCNN—deep cascaded neural network; LSTM—long short-term memory.

**Table 2 sensors-24-02097-t002:** Finite state machine (FSM) inputs.

Input	Description
00	No obstacles detected, indicating a clear path ahead.
01	An obstacle is being tracked, requiring speed adjustments to maintain safe following distances.
10	A red traffic light is ahead, necessitating a controlled stop.
11	A stop sign is detected, also demanding a full stop.

**Table 3 sensors-24-02097-t003:** State transition table (based on hand-crafted rules).

Present State	Next State				Output/Description
	Input = 00	Input = 01	Input = 10	Input = 11	
S1	S1	S2	S3	S4	DR/Drive
S2	S1	S2	S3	S4	FL/Follow the Leader
S3	S1	S2	S3	S4	ST/Stop Red Traffic Light
S4	S1	S2	S3	S4	SS/Stop Sign

**Table 4 sensors-24-02097-t004:** Non-linear model predictive control (MPC) parameters.

Δt	*H*	$C_{x}$	$C_{y}$	$C_{\theta }$	$C_{\kappa }$	$C_{\tau }$
1 s	4 s	5	5	10	100	10

**Table 5 sensors-24-02097-t005:** Time execution for main modules in hybrid and modular architectures.

Module	Module/Algorthm	Inputs	Outputs	Architecture	Average Execution Time (ms)	Device
Perception/3D detection	PointPillars	Point cloud	obstacles 3D	Modular/Hybrid	95	GPU
Perception/2D detection	Mask-RCNN	RGB image	detections 2D	Modular/Hybrid	198	GPU
Perception/Depth	ELAS	RGB Images	obstacles 3D	Modular/Hybrid	210	CPU
Planning/Path planning	CNN-Planner	BEVSFusion	local waypoints	Hybrid	5	GPU
Planning/local planner	map	global waypoints	local waypoints	Modular	0.61	CPU
Control/Lateral	MPC	local waypoints, pose	steering angle	Modular/Hybrid	0.5	CPU
Control/Longitudinal	PID	speed reference	break, throttle	Modular/Hybrid	0.1	CPU
Decision-Making	FSM	Obstacles in the path	speed reference	Modular/Hybrid	0.4	CPU

**Table 7 sensors-24-02097-t007:** Results: CARLA challenge 2023. CARLA leaderboard 2, Track MAP.

Team	Method	DS	RC	IP	CP	CV	CL	RLI	SSI	ORI	RD	RT	AB	YEI	ST	MinSI
Kyber-E2E	Kyber-E2E	3.11	5.28	0.67	0.36	0.63	0.27	0.09	0.09	0.01	0.00	0.09	0.09	0.00	0.54	0.00
**LRM 2023**	**CaRINA agent**	1.14	3.65	0.46	0.00	2.89	1.31	0.00	0.53	0.00	0.13	1.31	1.18	0.00	2.10	0.00

**Table 8 sensors-24-02097-t008:** Results: CARLA Leaderboard 1, Track SENSORS.

Team	Method	DS	RC	IP	CP	CV	CL	RLI	SSI	ORI	RD	RT	AB
Interfuser	ReasonNet [15]	79.95	89.89	0.89	0.02	0.13	0.01	0.08	0.00	0.04	0.00	0.01	0.33
Interfuser	InterFuser [16]	76.18	88.23	0.84	0.04	0.37	0.14	0.22	0.00	0.13	0.00	0.01	0.43
PPX	TCP [17]	75.14	85.63	0.87	0.00	0.32	0.00	0.09	0.00	0.04	0.00	0.00	0.54
DP	TF++ WP Ensemble [58]	66.32	78.57	0.84	0.00	0.50	0.00	0.01	0.00	0.12	0.00	0.00	0.71
WOR	LAV [69]	61.85	94.46	0.64	0.04	0.70	0.02	0.17	0.00	0.25	0.09	0.04	0.10
Attention Fields	TF++ WP [58]	61.57	77.66	0.81	0.02	0.41	0.00	0.03	0.00	0.08	0.00	0.00	0.71
DP	TransFuser [65,70]	61.18	86.69	0.71	0.04	0.81	0.01	0.05	0.00	0.23	0.00	0.01	0.43
Attention Fields	Latent TransFuser [65,70]	45.20	66.31	0.72	0.02	1.11	0.02	0.05	0.00	0.16	0.00	0.04	1.82
RaphaeL	GRIAD [59]	36.79	61.85	0.60	0.00	2.77	0.41	0.48	0.00	1.39	1.11	0.34	0.84
**LRM 2023**	**CaRINA hybrid**	35.36	85.01	0.45	0.02	4.95	0.22	1.67	0.12	0.45	1.54	0.02	0.45
WOR	World on Rails [62]	31.37	57.65	0.56	0.61	1.35	1.02	0.79	0.00	0.96	1.69	0.00	0.47
MaRLn	MaRLn [63]	24.98	46.97	0.52	0.00	2.33	2.47	0.55	0.00	1.82	1.44	0.79	0.94
Attention Fields	NEAT [64]	21.83	41.71	0.65	0.04	0.74	0.62	0.70	0.00	2.68	0.00	0.00	5.22
SDV	AIM-MT [64]	19.38	67.02	0.39	0.18	1.53	0.12	1.55	0.00	0.35	0.00	0.01	2.11
SDV	TransFuser (CVPR 2021) [65]	16.93	51.82	0.42	0.91	1.09	0.19	1.26	0.00	0.57	0.00	0.01	1.96
LRM-B	CNN-Planner [66]	15.40	50.05	0.41	0.08	4.67	0.42	0.35	0.00	2.78	0.12	0.00	4.63
LBC	Learning by Cheating [67]	8.94	17.54	0.73	0.00	0.40	1.16	0.71	0.00	1.52	0.03	0.00	4.69
Attention Fields	CILRS [68]	5.37	14.40	0.55	2.69	1.48	2.35	1.62	0.00	4.55	4.14	0.00	4.28
LRM 2019	CaRINA [10]	4.56	23.80	0.41	0.01	7.56	51.52	20.64	0.00	14.32	0.00	0.00	10,055.99

**Table 9 sensors-24-02097-t009:** Results: CARLA challenge 2023. CARLA leaderboard 2, Track SENSORS.

Team	Method	DS	RC	IP	CP	CV	CL	RLI	SSI	ORI	RD	RT	AB	YEI	ST	MinSI
**LRM 2023**	**CaRINA hybrid**	**1.23**	9.55	0.31	0.25	**1.64**	**0.25**	0.25	**0.40**	0.43	0.10	0.30	0.60	0.10	1.20	**0.15**
Tuebingen_AI	Zero-shot TF++ [58]	0.58	8.53	0.38	0.17	1.80	0.51	0.00	3.76	0.35	0.06	0.56	0.51	0.00	2.19	0.17
CARLA	baseline	0.25	15.20	0.10	1.23	2.49	0.79	0.03	0.94	0.47	0.50	0.00	0.13	0.13	0.69	0.19

## Data Availability

Data are contained within the article.

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
