# Peer review of "Integrating Modular Pipelines with End-to-End Learning: A Hybrid Approach for Robust and Reliable Autonomous Driving Systems"

_sensors, 2024, doi:10.3390/s24072097_

Round 1

Reviewer 1 Report

Comments and Suggestions for Authors

The abstract mentions the integration of modular and end-to-end approaches but lacks specific results to quantify improvements over existing methods. A recommendation is to include key performance metrics that highlight the hybrid system’s advancements.

The introduction could better frame the research gap by specifically comparing the limitations of purely modular versus end-to-end approaches that the hybrid model aims to overcome. It is suggested to add a comparative analysis or a table summarizing these aspects.

It is mentioned that sensor fusion enhances perception accuracy, yet there is no comparative analysis or specific data showing the improvement over non-fusion methods. It is suggested to provide quantitative results or a case study that illustrates the impact of sensor fusion on perception reliability.

The results section showcases performance in the CARLA challenge but does not compare these outcomes directly with baseline systems or previous approaches. It would be better to include a detailed table comparing this hybrid system’s performance metrics (like accuracy, processing time, etc.) against those of both purely modular and end-to-end systems in similar conditions.

While the discussion acknowledges areas for improvement, it lacks depth in analyzing why certain challenges, such as traffic light detection and collision avoidance, were more problematic. It is recommended to provide a deeper analysis of these issues, possibly including failure case studies, which would offer insights into potential solutions or areas for future research.

Comments on the Quality of English Language

Minor editing of English language required.

Author Response

Please see the attached text. 

Reviewer 2 Report

Comments and Suggestions for Authors

1. The paper mentions the challenge of extensive coupling in modular architectures. Elaborate on how this challenge was addressed in the proposed hybrid approach, and provide insights into the system’s adaptability and robustness in the face of potential errors?

2. In Section 3, along with the modular software architecture discussion, it would be beneficial to include information on the algorithms used for localization, object detection, tracking, prediction, and decision-making. This would provide a more detailed understanding of the individual components.

3. When discussing challenges in the modular architecture (lines 101-106), could you elaborate on potential solutions or mitigations proposed in literature or applied in the presented hybrid approach to overcome issues related to error propagation and component management?

4. In Section 2.3, along with the hybrid architecture discussion, it would be helpful to elaborate on the specifics of the early-fusion approach mentioned. How does the fusion of LiDAR Bird’s Eye View images, Stereo Camera projections, and high-level commands enhance the spatial and semantic representation in data-driven path planning?

5. The paper mentions a “Prediction” approach for safe navigation, but the details provided are minimal. Elaborate on the specific algorithms or methods used for predicting the movements of surrounding objects? How accurate and computationally efficient is the chosen prediction model?

6. How is the model trained, and how does it handle different environmental conditions and lighting variations?

7. In section 3.1.2, Path Planning, the text mentions “sparse waypoints”. Elaborate on how these sparse waypoints are generated or selected, and what considerations are taken into account?

8. Elaborate on the choice of the ELAS algorithm for generating 3D point clouds from stereo images? What specific advantages or characteristics of ELAS make it suitable for your application?

9. The LiDAR and stereo camera processing pipeline is well-explained. However, discuss any challenges or limitations encountered in transforming LiDAR point clouds to the BEV coordinate system? How robust is this transformation in handling various environmental conditions?

10. How well does the proposed hybrid CaRINA agent generalize to unseen environments or scenarios? Are there any specific scenarios where the agent might face challenges or limitations?

Author Response

Please see the attached text. 

Reviewer 3 Report

Comments and Suggestions for Authors

The manuscript is devoted to the development of an autonomous driving navigation system. The authors consider various aspects of the system, in particular, sensing, obstacle detection, risk assessment, decision-making, planning and control management. The authors proposed a hybrid architecture that combines modular perception and control modules with data-driven path planning. An analysis of the proposed approach and a comparison with other existing solutions are presented. In general the manuscript looks convincing and good study was conducted. I have the following remarks.

1) The importance of real time is declared several times in the text. However, the manuscript does not provide data on the computational time and efficiency of the methods. So it remains unknown whether the goal (real time) has been achieved or not.

2) The abbreviations should be revealed in the first place where they appear (for example, LRM on the line 194, FSM on the line 296, MPC on the line 360, GNSS on the line 430 and more). Also, the ELAS algorithm is mentioned in Fig. 4 and on the line 487 but the appropriate reference is only added on the line 524.

3) Fig. 4 presents the perception module and contains the Disparity Depth block. But nothing is said about this block in section 3.2 (Perception). A block description should be added.

4) Table 3 and Fig. 6 look unnecessary. The only useful info they demonstrate in complicated form is that the next state does not depend on the current state, but only on input presented in Table 2. It is much easier to read this info in one sentence than to comprehend it from Fig.6.

5) Please remove “w.r.t.” in lines 435, 440 and 447.

6) Line 668: “We also obtained a high driving score (DS=35.36) compared to similar approaches.” Please list these approaches here explicitly.

7) Not all references are formed according to the requirements.

8) Line 373: “Target speed is set to a maximum of 8.8 m/s (31.68 km/h).” Why is exactly 31.68km/h the maximum speed?

Comments on the Quality of English Language

Some minor correction of English is required, for example, mistyping “withe” (line 487) or extra point (line 620).

Author Response

Please see the attached text. 

Round 2

Reviewer 1 Report

Comments and Suggestions for Authors

All my comments were addressed and I have no more comments.

Reviewer 2 Report

Comments and Suggestions for Authors

Revised version looks fine and no more comments.